# Assessment of Standard Surgical Excision Efficacy and Analysis of Recurrence-Associated Factors in 343 Cases of Nasal Basal Cell Carcinoma: A Single-Center Retrospective Study

**DOI:** 10.3390/healthcare12050513

**Published:** 2024-02-21

**Authors:** Karolina Baltrušaitytė, Ernest Zacharevskij, Loreta Pilipaitytė, Kęstutis Braziulis, Arūnas Petkevičius

**Affiliations:** 1Faculty of Medicine, Lithuanian University of Health Sciences, 44307 Kaunas, Lithuania; 2Department of Plastic and Reconstructive Surgery, Hospital of Lithuanian University of Health Sciences Kaunas Clinics, 50161 Kaunas, Lithuania; ernest.zacharevskij@gmail.com (E.Z.); loreta.pilipaityte@kaunoklinikos.lt (L.P.); kestutis.braziulis@kaunoklinikos.lt (K.B.); 3Department of Dermatovenerology, Hospital of Lithuanian University of Health Sciences Kaunas Clinics, 50161 Kaunas, Lithuania; dermatologas@gmail.com

**Keywords:** basal cell carcinoma, non-melanoma skin cancer, recurrence, standard surgical excision

## Abstract

In Caucasians, basal cell carcinoma, the predominant non-melanoma skin cancer type, poses challenges for surgeons due to anatomical and aesthetic concerns, particularly when located on the nose. The study aimed to evaluate tumor distribution, size, morphological subtypes, surgical outcomes, radicality levels, and their correlation with recurrence rates. A retrospective analysis encompassed 343 cases of nasal skin cancer over a four-year period from 1 January 2019 to 31 December 2022. The research cohort comprised 252 female and 91 male participants, averaging 75.2 years old. Tumors were most found on the left sidewall of the nose (25.4%) and the dorsum (24.8%). The infiltrative morphological subtype was predominant (70.8%). Standard surgical excision with fasciocutaneous plastic was the preferred surgical procedure. Radical excision, defined by the absence of tumor cells in a resection margin, was accomplished in 79.0% of lesions, whereas 16.9% demonstrated incomplete excision, signifying the presence of tumor cells in the resection margin. Non-radically excised tumors exhibited a significantly higher recurrence rate (24.1%) compared to those with radical excision (6.3%). In nasal reconstruction, diverse surgical techniques are essential for precise adaptation based on factors like tumor characteristics and patient needs. Despite surgeons’ careful adherence to excision margin guidelines, the possibility of non-radical outcome cannot be eliminated.

## 1. Introduction

The British Association of Dermatologists states that basal cell carcinoma (BCC) is the predominant malignancy among non-melanoma skin cancers in Caucasians, arising from epidermal cell origins. Its occurrence in the nasal region presents intricate challenges for surgeons due to anatomical, functional, and aesthetic issues [1,2,3,4]. The prevalence of this heterogeneous group of tumors, encompassing a spectrum of histopathological and clinical traits spanning from superficial lesions to extensive, destructive manifestations, has demonstrated a consistent upward trend on a global scale in recent decades, notably among the elderly population [1,5,6,7]. In European countries, including Lithuania, the average life expectancy of the population is on the rise. This demographic shift is accompanied by an annual increase of approximately 5% in the incidence of BCC cases, accounting for 80% of the total prevalence in the facial region, with 31.3% specifically located on the nose [3,4,8,9]. The tumor’s geographic distribution varies, with Australia reporting the highest incidence rates (726 to 1000 cases per 100,000 person-years) and Africa the lowest (fewer than one case per 100,000 person-years). In European countries, including Ireland, Great Britain, Finland, Lithuania, Germany, Switzerland, Italy, France, and Spain, incidence rates range from 44.6 to 157 cases per 100,000 individuals annually [4,10]. As a result, early diagnosis and effective treatment of such tumors are becoming increasingly pertinent in the routine clinical practice of physicians. According to the National Comprehensive Cancer Network (NCCN) guidelines, basal cell carcinoma is stratified into low- and high-risk tumors, as well as locally advanced and metastatic disease, contingent on histopathological subtypes and diverse anatomical localizations, resulting in distinct treatment approaches [11,12]. In this investigation conducted at the Department of Plastic and Reconstructive Surgery, Hospital of Lithuanian University of Health Sciences, we examine the surgical management of nasal basal cell carcinoma, assessing the attained level of surgical radicality and subsequently exploring potential correlations between radicality and recurrence rates. Hence, the primary treatment methods adopted in our department may deviate from the guidelines proposed by the British Association of Dermatologists or other recommendations for dermatologists [4,13,14]. Concentrating specifically on the outcomes of surgical intervention, our investigation reveals that recent literature underscores the association between incomplete tumor excision and an augmented risk of recurrence [6,12,15,16]. These rates fluctuate among different studies, with non-radical surgery occurring in the range of 6% to 50% and recurrence rates ranging from 3% to 38%. Notably, surgical excision of BCC on the nose exhibited a recurrence rate 2.5 times higher than that observed in skin cancer at other anatomical sites [8,17,18,19]. Indeed, given that re-excision of BCC and subsequent treatment demands more intricate surgical considerations and entails higher costs, this amplifies the burden on both the healthcare system and the affected individuals. Consequently, the successful treatment of BCC continues to be a crucial healthcare concern [13]. 

## 2. Materials and Methods

A retrospective study was conducted on patients with basal cell carcinoma (BCC) in the nasal region at the Department of Plastic and Reconstructive Surgery, Hospital of Lithuanian University of Health Sciences Kaunas Clinics. Ethical approval for this study was obtained from Kaunas Regional Biomedical Research Ethics Committee (16 December 2022, No. BEC-MF-123). The scientific research focused on patients who underwent standard surgical excision of the nose for basal cell carcinoma (BCC) January 2019, and 31 December 2022. Standard surgical excision (SSE) adhered to the National Comprehensive Cancer Network (NCCN) guidelines, encompassing a 4 mm clinical margin for low-risk BCCs and 5 mm for high-risk BCCs. Resected BCC specimens underwent histological examination to assess resection margins and determine radicality—defined as the absence of tumor cells in margins. The application of the Mohs micrographic technique was excluded from our study due to financial constraints and the requisite additional equipment for surgical treatment. The study involved the analysis of patients’ medical records, which were accessed through the Hygiene Institute Health Information Centre, utilizing the International Classification of Diseases, Tenth Revision, Clinical Modification (ICD-10-CM) code: C44.3. After a comprehensive review of 572 case histories, 343 eligible patients were identified and included in the study. Patients were only included if they met all the study’s inclusion criteria: they were older than 18 years old during inclusion in the study, possessed a confirmed diagnosis of basal cell carcinoma confined to the nasal region, as validated by the ICD-10-AM code: C44.3. Additionally, they were mandated to provide informed written consent prior to their participation in the research. The exclusion criteria precluded individuals falling within the following categories: those below 18 years of age, patients with incomplete medical records lacking precise information regarding the localization, size, or type of surgical intervention for basal cell carcinoma, and individuals who expressed dissent towards participation in the research. Throughout this study, subject confidentiality was meticulously safeguarded. All patient health data were anonymized and encoded to prevent direct identification. Dissemination of medical information to third parties or institutions only occurred in compliance with legal mandates. Retrospectively, information pertaining to age, gender, mean size of the basal cell carcinoma in the nasal region, surgical modality, histopathological assessments, surgical radicality, and recurrence frequency in the study cohort was gathered. Subsequently, the acquired data underwent statistical analysis, enabling comparisons and the elucidation of the aforementioned criteria integral to the research evaluation. The data were collected and organized utilizing Microsoft Office Excel 2019. Statistical analyses were conducted using the IBM Statistical Package for Social Sciences (SPSS) 27.0 software. The normality assumption of continuous variables was assessed employing the Kolmogorov–Smirnov test. In the realm of analytical statistics, for bivariate data analyses, the chi-square (χ^2^) criterion, Student’s *t*-test (applied for normally distributed variables), and Mann–Whitney U test (utilized for non-normally distributed variables) were employed to assess the distribution of phenomena across groups. Normally distributed data were expressed as mean ± standard deviation, while non-normally distributed data were presented as median or as both median and mean in the event of identical medians. Statistically significant disparities and associations between features were considered valid when the calculated *p*-value was less than the predetermined significance level (α = 0.05).

## 3. Results

Throughout the course of this research endeavor, a total of 572 surgical interventions were conducted for the removal of basal cell carcinoma in the nasal region. Of these, 343 patients were deemed to meet the established inclusion criteria, comprising 252 females (73.5%) and 91 males (26.5%). In this heterogeneous cohort, there was a statistically significant predominance of females over males (*p* < 0.001, χ^2^ = 75.571). The mean age of the participating patients was 75.18 ± 10.22 years, with the youngest individual being 43 years old and the eldest being 98 years old.

### 3.1. Characteristics of Nasal Basal Cell Carcinoma

The research aimed to investigate the distribution of BCC localization within the nasal region. An observable trend revealed a statistically significant preponderance of tumor formations in regions that were directly exposed to sunlight, particularly in Zone H, which encompasses the left sidewall of the nose and the dorsum. These areas accounted for 87 cases (25.4%) and 85 cases (24.8%), respectively. Conversely, the least frequent occurrence of BCC was noted on the tip of the nose, with only 40 cases (11.7%) identified. The observed differences in BCC localization were found to be highly significant (*p* < 0.001, χ^2^ = 23.166) (Table 1).

Another characteristic parameter of BCC examined in this study was tumor size. To facilitate a precise evaluation of tumor diameter, subjects were stratified by gender, and the measurement of this parameter was conducted separately for each group. Analysis of the results (Table 2) revealed that there was no statistically significant difference in BCC size between female and male patients (0.79 ± 0.38 cm and 0.82 ± 0.44 cm, respectively) (*p* > 0.05). The smallest recorded tumor diameter in females was 0.2 cm, whereas in males it was 0.3 cm. The largest observed diameters were 2.5 cm in females and 3.5 cm in males.

According to the presented data (Table 3), the infiltrative histopathological subtype of BCC was the most prevalent, observed in 243 cases (70.8%), whereas the sclerosing morphological subtype was the least common, occurring in only 13 cases (3.8%). The application of the goodness-of-fit chi-square test revealed a statistically significant disparity in the distribution of data within the group, indicating heterogeneity among the groups (*p* < 0.001, χ^2^ = 565.79).

### 3.2. Surgical Management of Nasal Basal Cell Carcinoma: Methods, Findings, and Long-Term Outcomes

Following the statistical examination of the dataset, the prevalence of selected surgical treatment modalities among the sampled patients was computed. The findings of this study reveal that, for the excision of BCC in various nasal regions, the most employed surgical approach was standard surgical excision (SSE) with fasciocutaneous plastic, implemented in 262 cases (76.4%). Conversely, the least frequently employed technique was excision coupled with nasal reconstruction, utilized in only six cases (1.7%) (*p* < 0.001, χ^2^ = 306.9).

This study also aimed to ascertain the level of radicality in the surgical management of patients with BCC and to assess its correlation with recurrence frequency. According to the data of the scheme (Figure 1), 271 cases (79.0%) underwent radical excisions, while 58 cases (16.9%) were subjected to non-radical removal. A notable association was observed, wherein non-radical excisions exhibited a higher recurrence rate (24.1%) compared to radical excisions, which demonstrated a recurrence rate of only 6.3% (*p* < 0.001, χ^2^ = 18.108). Subsequent evaluation of the influence of variables such as gender, tumor size, and morphological subtype on BCC recurrence did not yield statistically significant differences (*p* > 0.05).

Upon analyzing the selection of subsequent treatment strategies for patients, considering the radicality of the initial operation, it was observed that in cases where tumors were non-radically removed (R1), further treatment involved a recommendation for re-excision when recurrence becomes apparent, constituting 23 cases (39.7%). Additionally, re-excision within 0–6 months was pursued in 17 cases (29.3%) (*p* < 0.001, χ^2^ = 310.99). Comparatively, the less frequently employed strategies encompassed re-excision after more than 12 months and targeted re-excision, accounting for 8.6% and 10.3% respectively. The term "targeted re-excision" refers to the recommendation by the plastic surgeon for a follow-up excision of basal cell carcinoma. However, the patient did not comply with the recommendation and did not undergo the recommended targeted re-excision procedure. Within the category of radically excised tumor formations (R0), a statistically significant distribution of data was observed. Remarkably, the predominant approach applied to the vast majority of subjects, encompassing 92.6%, involved a dermatologist’s consultation after a span of 2–3 months. In this group, repeated surgical treatment was rarely necessary, occurring in only 0.7% of cases (*p* < 0.001, χ^2^ = 310.99) (Table 4).

In the conducted data analysis, the frequency distribution of surgical radicality in patients undergoing treatment for BCC in the nasal region at the Department of Plastic and Reconstructive Surgery, Hospital of Lithuanian University of Health Sciences Kaunas Clinics, was assessed from 2019 to 2022. As depicted in Figure 2, a discernible trend was noted, with the proportion of radical (R0) excisions increasing from 73.8% in 2019 to 81.2% in 2022. Conversely, the incidence of incompletely removed tumor formations (R1) exhibited an inverse trajectory, decreasing from 20.6% in 2019 to 15.7% in 2022. Following the application of the non-parametric chi-square test for independence of attributes in the statistical analysis, it was determined that the distribution of data within the R1 group exhibited a statistically significant difference, primarily stemming from a notable reduction in the number of R1 excisions over the course of these four years (Figure 2).

## 4. Discussion

Lultschik et al. conducted a retrospective study in Canada in 2023 to examine the distribution of basal cell carcinoma based on age, gender, and recurrence rates. The study sample comprised 155 (55.8%) females and 123 (44.2%) males, with an average participant age of 57.2 years. The statistical analysis revealed that the results did not exhibit significant differences in the age and gender groups [20]. While the distribution of patients by gender aligns with our study, a notable disparity emerged in the evaluation of age metrics. In our research, participants were notably older, with an average age differing by approximately 18 years compared to the referenced study. Devine et al. (2017) similarly identified a younger sample in Great Britain, with a mean age of 72.5 years [21]. We also reviewed other studies where, in contrast to our data, the majority of subjects were male [3,22,23,24]. Ghanadan et al. (2014) attributed this gender distribution to factors such as women’s higher usage of products containing SPF (sun protection factor) for face protection from ultraviolet (UV) rays and men’s increased outdoor exposure leading to greater direct UV radiation exposure [25]. The predominant representation of women in our study can be elucidated by the admission process, where patients transition to the Department of Plastic and Reconstructive Surgery from the Department of Dermatology without specific gender-based selection during the study. The observed gender disparity is ascribed to the elevated health-conscious behavior of women, leading to more frequent medical consultations for the examination of tumor formations and subsequent treatment. This behavior contributes to the higher participation of women in our study.

In a 2019 retrospective cohort single-center study authored by De Nicolo et al. and published in Italy, the primary focus centered on the comprehensive evaluation of histopathological classifications, anatomical localization, and size distribution of BCC within facial regions. The study’s objective was to delineate potential correlations between these attributes and the likelihood of recurrence. The findings exhibited a preponderance of tumors predominantly located in the nasal region (31.8%), with 11.4% situated on the dorsum and an equivalent 8.0% present on both the right and left nasal sidewalls. The prevalent histopathological manifestations of BCC were noted as nodular and sclerodermiform subtypes, accounting for 65.9% and 15.9%, respectively. The calculated average dimensions of the tumors, determined through the length-to-width ratio, amounted to 0.9 × 0.7 cm, yielding an area of 0.83 cm^2^ [16]. In comparison to our own study, notable distinctions emerged. Specifically, our investigation identified the primary site of BCC occurrence to be the left nasal sidewall (25.4%). Furthermore, the nodular morphological subtype ranked second in prevalence, representing only 12.8%. In stark contrast, the infiltrative subtype prevailed as the most common histological manifestation, encompassing a substantial 70.8%. The prevalence of the infiltrative morphological subtype in our study is predominantly attributable to the study participants being referred for surgical treatment to plastic and reconstructive surgery specialists from the Department of Dermatology. In this setting, cases of basal cell carcinoma with a superficial presentation are treated locally using topical medication such as imiquimod 5% ointments, while more intricate cases, often exhibiting infiltrative subtype, are directed for more aggressive surgical treatment following biopsy procedures. This referral and treatment approach contributed to the observed prevalence of the infiltrative morphological subtype in our study. Additionally, our assessment of average tumor size was predicated exclusively on the mean diameter dimensions, resulting in a smaller measurement of 0.8 cm, in contrast to the reported analysis. It is essential to acknowledge that De Nicolo et al. did not report any instances of BCC recurrence, underscoring the necessity to consult additional scientific sources for a more comprehensive evaluation of the interplay between these attributes and recurrence rates. Morgan et al. conducted a comprehensive analysis of 496 cases of BCC, stratifying their cohort based on tumor size, distinguishing between those exceeding and falling below a 2 cm diameter threshold. The investigation determined that larger tumors, surpassing 2 cm, exhibited a notably heightened recurrence risk, quantified at 8.9%, in stark contrast to the 0.8% recurrence rate observed in tumors measuring less than 2 cm [26]. Additionally, in a study undertaken by Kondo et al., the more aggressive micronodular and sclerodermiform subtypes accounted for 17.24% and demonstrated a statistically significant association with recurrence compared to other subtypes (*p* = 0.0001) [27]. Based on our study’s findings, it is reasonable to infer that tumor attributes such as anatomical localization, morphological classification, and size exerted negligible influence on the occurrence of BCC recurrences. This inference is drawn from the absence of a discernible statistical correlation between the recurrence rate and these specific characteristics.

An analysis was conducted on a prospective randomized controlled trial by Kofler et al. in Germany in 2021. The study compared the radicality and recurrence rate between surgical excision with serial section histology and 3D-histology in the management of facial basal cell carcinoma. This investigation encompassed 569 BCC cases, comprising 59 subjects (20.9%) who underwent serial section histology and 113 (39.4%) who underwent 3D-histology. The statistical findings from this study reveal that the frequency of non-radical operations was 30.2%, and the recurrence rate with serial section histology stood at 8.4%. In contrast, the use of 3D-histology yielded a significantly lower recurrence rate of 3.5% among subjects, resulting in an overall recurrence frequency of 6.0% [28]. As our study is conducted from the perspective of plastic and reconstructive surgery, the criteria for successful surgical treatment and their comparison with data from other authors are of particular significance. Therefore, we opted to compare the distribution of non-radical operations and the frequency of recurrence. In our own study, which exclusively employed standard surgical excisions with serial section histology, the incidence of non-radical operations was notably lower compared to the aforementioned study (16.9%). Nevertheless, it is worth noting that scientific literature suggests that this indicator generally falls within the range of 1 to 40% worldwide [19,29,30,31,32]. To obtain a more precise estimation of the prevalence of incompletely removed tumors, we conducted a literature review. Wollina et al. (2014) performed a retrospective study in Germany and found that the rate of non-radical (R1) excisions was 30.5% in BCC treated with SSE [8]. Szewczyk et al. (2022) reported a frequency of 19.6% for R1 excisions [22]. In a retrospective study by Ürün et al. (2022), they found that the rate of non-radical (R1) excisions was 18.9%, and the recurrence rate was 9.6% in BCC treated with SSE. Their findings also demonstrated a statistically significant higher recurrence rate for incompletely removed tumors compared to those removed with radical excision (*p* < 0.05) [5]. In our research, a similar trend of statistical dependence was observed, with BCC recurrence after R1 operations being 14.5% higher, while the overall recurrence was 3.2% lower. However, in other studies, this parameter exhibits a wider range of variation (25.0–46.0%) [32]. We also explored publications with lower percentages for the analyzed criteria. Veronese et al. (2012) reported that non-radical surgery in their study accounted for only 7.2%, of which 4.2% recurred, while radically removed tumors recurred in only 3.4% of the study participants. The overall recurrence rate was 8.3% [33]. According to the results of our study, the overall recurrence rate of BCC in the nasal area (6.4%) possibly increased because the majority of the study sample consisted of the aggressive and, according to literature sources, more prone to relapse infiltrative BCC morphological subtype [3,9,16,25,26].

This research paper presents a four-year analysis of surgical treatment for nasal basal cell carcinoma, focusing on radicality and recurrence rates. The study underscores the importance of enhancing early diagnosis and comprehensive examinations to detect tumors at earlier stages, potentially improving surgical outcomes and reducing treatment costs. With a rising demographic trend in BCC cases, consideration should be given to adopting the Mohs micrographic technique at the Hospital of Lithuanian University of Health Sciences Kaunas Clinics Department of Plastic and Reconstructive Surgery, known for reducing non-radical operations and achieving high cure rates. Emphasis on determining surgical margins during procedures and adhering to NCCN guidelines, maintaining margins of 4 mm for low-risk and 5 mm for high-risk basal cell carcinomas, is crucial, especially when opting for more intricate surgical techniques. The study suggests the need for extended investigations with larger patient samples to compare diverse operative treatment methods and obtain more precise data on their respective advantages. Nonetheless, the research paper exhibits several methodological limitations. Primarily, a retrospective study design was chosen, relying on data extracted from patients’ medical records. The assessment and documentation of objective symptoms are inherently subjective, potentially introducing inaccuracies during re-evaluation. It is essential to acknowledge that this scientific investigation was confined to a single medical institution, which may impact the overall accuracy of the data, especially if the studied patients sought care in other healthcare facilities or refrained from consulting at all due to recurrent skin oncological conditions outside the scope of this study. Additionally, when comparing our data analysis with international studies, it becomes evident that the research sample size was relatively limited, and certain surgical treatment methods considered as the gold standard in other countries are not available in Lithuania due to financial constraints and complex techniques. Consequently, it is imperative to acknowledge the constraints of this research, potential sources of bias, and to approach the results with circumspection.

## 5. Conclusions

The findings derived from this study necessitate cautious interpretation and call for substantiation through subsequent research efforts. Nonetheless, they suggest a predominant focus on surgical intervention for basal cell carcinoma (BCC) among elderly female patients. The primary tumor localization exhibited notable prominence on the left sidewall of the nose and the dorsum. Additionally, there was no statistically significant disparity in the size of BCC between the genders, with the infiltrative morphological type being the most prevalent. The preferred surgical modality entailed standard surgical excision with fasciocutaneous plastic reconstruction. Radical excisions of BCC, surpassing non-radical procedures by a factor of four, exhibited an association with reduced recurrence rates. It is deduced that nasal reconstruction mandates a comprehensive repertoire of surgical techniques to precisely tailor the procedures in accordance with factors such as tumor localization, size, depth, and the specific requirements of individual patients. Despite the meticulous adherence of surgeons to excision margin guidelines for basal cell carcinoma, the possibility of non-radical outcome cannot be entirely avoided.

## Figures and Tables

**Figure 1 healthcare-12-00513-f001:**
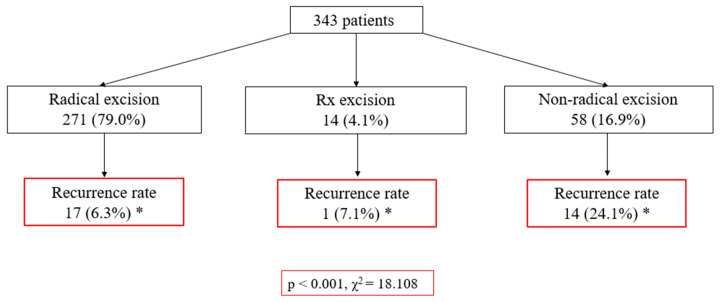
Patient distribution following surgical treatment of basal cell carcinoma based on radicality and recurrence rate. Note: * statistically significant value *p* < 0.05; Cells highlighted in red are considered statistically significant; Rx excision—excision of undetermined extent.

**Figure 2 healthcare-12-00513-f002:**
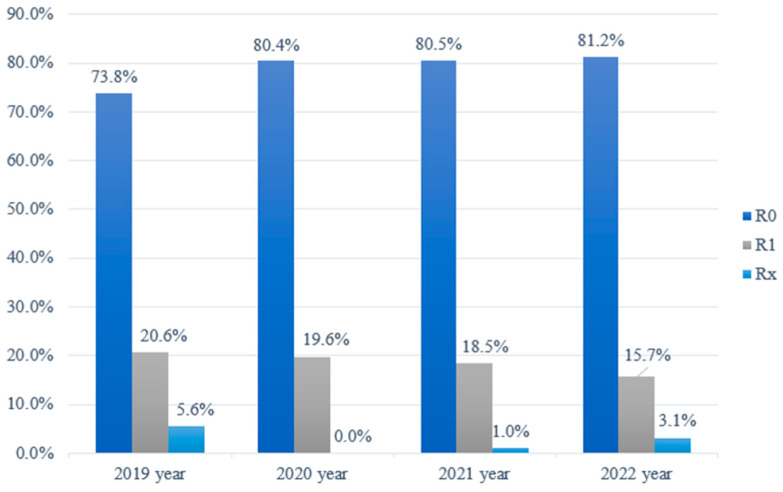
Patient distribution following surgical treatment of basal cell carcinoma based on surgical radicality from 2019 to 2022.

**Table 1 healthcare-12-00513-t001:** Patient distribution by localization of basal cell carcinoma.

Localization of Basal Cell Carcinoma	Number	Percentage	*p*-Value
Left sidewall	87	25.4	<0.001 χ^2^ = 23.166
Dorsum	85	24.8	
There is no exact localization	74	21.5	
Right sidewall	57	16.6	
Tip	40	11.7	

**Table 2 healthcare-12-00513-t002:** Patient distribution by tumor size of basal cell carcinoma.

Gender	Mean ± SD of Tumor Size (cm)	Median (min–max) (cm)	*p*-Value
Female	0.79 ± 0.38	0.7 (0.2–2.5)	0.481
Male	0.82 ± 0.44	0.8 (0.3–3.5)	

**Table 3 healthcare-12-00513-t003:** Patient distribution by histopathological subtype of basal cell carcinoma.

Histopathological Subtype	Number	Percentage	*p*-Value
Infiltrative	243	70.8	<0.001 χ^2^ = 565.79
Nodular	44	12.8	
Superficial	33	9.6	
Sclerosing	13	3.8	
Not indicated	10	2.9	

**Table 4 healthcare-12-00513-t004:** Patient stratification following surgical treatment of basal cell carcinoma based on further treatment approach and surgical radicality.

Radicality	Dermatologist Consultation after 2–3 Months	Re-Excision after 0–6 Months	Re-Excision after >12 Months	Re-Excision When Recurrence Becomes Apparent	Targeted Re-Excision	*p*-Value
Radically excised tumors	251 (92.6%)	0	2 (0.7%)	0	0	<0.001 χ^2^ = 310.99
Non-radically excised tumors	0	17 (29.3%)	5 (8.6%)	23 (39.7%)	6 (10.3%)	

## Data Availability

The data presented in this study are available in article.

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
