# Peer review of "Assessment of Standard Surgical Excision Efficacy and Analysis of Recurrence-Associated Factors in 343 Cases of Nasal Basal Cell Carcinoma: A Single-Center Retrospective Study"

_healthcare, 2024, doi:10.3390/healthcare12050513_

Round 1
Reviewer 1 Report
Comments and Suggestions for Authors
Assessment of Standard Surgical Excision Efficacy and Analy-2 sis of Recurrence-Associated Factors in 343 Cases of Nasal Basal 3 Cell Carcinoma: a single-center retrospective study
Dear Authors,
I have read with interest your manuscript. Next, I will proceed to explain my doubts and concerns about this paper:
Abstract
The studied cohort of patients has a clear female predominance, do you have a reason for that? Normally, CBC affects sex in similar frequency.
Why the infiltrative subtype was the predominant histological subtype?
The revision comprised recent years, justify why Mohs surgery was not indicated.
What is the difference between a radical excision and a not radical excision, you must describe that briefly.
Introduction
In my view it is too long, you should change it for introduce your research. Maybe you should state that the results are of a Plastic Surgery Service, and guidelines or management are different that the one proposed by the British Association of Dermatologist or others dermatologist associations, as dermatologist usually perform a biopsy for planning treatment and not always use surgery. The plastic surgeon as a different procedure for treatment.
Line 62 is not completely correct: The first line for an ease to treat CBC it is not a radical excision, less in the nose. It is one of the alternatives.
Material and methods
The same as in the abstract. You should define what is a surgical excision and why the histological analysis was assessed after the excision. You need to state the margins of the excision, it is very important, at least the margin planned during the surgical procedure.
Discussion
Authors should discuss their methods and their results and compare them with the literature. There is not a clear line of discussion in this. They should be deeper, why the sample as a predominant female percentage, why the location are like appears in your patients, why non-radical excision was selected, why the most predominant type was infiltrative despite been one of the less frequent.
And the most important, what this study add and what authors thing must be changed in the surgical management of BCC in the previous years.
What are visual guidelines?
Author Response
Thank you very much for taking the time to review this manuscript. Please find the detailed responses below and the corresponding corrections highlighted in the re-submitted files.
Comments 1:
The studied cohort of patients has a clear female predominance, do you have a reason for that? Normally, BCC affects sex in similar frequency.
Response 1:
Patients are admitted to the Department of Plastic and Reconstructive Surgery from the Department of Dermatology without methodological gender-based selection during the study. The observed disparity in gender representation is attributed to women's heightened health-conscious behavior, resulting in more frequent medical consultations for the examination of tumor formations and subsequent treatment, thereby contributing to the greater female participation in the study.
Comments 2:
Why the infiltrative subtype was the predominant histological subtype?
Response 2:
As previously stated, patients are transferred for surgical treatment from the Department of Dermatology. While milder (superficial) forms of BCC are locally treated with imiquimod ointments, more severe cases are directed to plastic surgeons for surgical intervention. Consequently, this led to a higher inclusion of patients with the infiltrative morphologic subtype in the study.
Comments 3:
The revision comprised recent years, justify why Mohs surgery was not indicated.
Response 3:
The exclusion of the Mohs surgical technique from the study is attributed to its substantial financial costs and the requisite specialized equipment, which regrettably are not currently available in Lithuanian hospitals, rendering this technique not yet implemented in our country.
Comments 4:
What is the difference between a radical excision and a not radical excision, you must describe that briefly.
Response 4:
In the abstract, we describe the distinction between radical and non-radical excision.
Comments 5:
In my view it is too long, you should change it for introduce your research. Maybe you should state that the results are of a Plastic Surgery Service, and guidelines or management are different that the one proposed by the British Association of Dermatologist or others dermatologist associations, as dermatologist usually perform a biopsy for planning treatment and not always use surgery. The plastic surgeon as a different procedure for treatment.
Response 5:
We have corrected the introduction according to your recommendation.
Comments 6:
Line 62 is not completely correct: The first line for an ease to treat CBC it is not a radical excision, less in the nose. It is one of the alternatives.
Response 6:
We addressed this issue by specifying that the study was conducted in the Department of Plastic and Reconstructive Surgery, emphasizing a focus on surgical treatment methods. This distinction implies potential deviations from dermatologists' recommendations in the treatment approach.
Comments 7:
The same as in the abstract. You should define what is a surgical excision and why the histological analysis was assessed after the excision. You need to state the margins of the excision, it is very important, at least the margin planned during the surgical procedure.
Response 7:
We have corrected this part according to your recommendation.
Comments 8:
Authors should discuss their methods and their results and compare them with the literature. There is not a clear line of discussion in this. They should be deeper, why the sample as a predominant female percentage, why the location are like appears in your patients, why non-radical excision was selected, why the most predominant type was infiltrative despite been one of the less frequent.
And the most important, what this study add and what authors thing must be changed in the surgical management of BCC in the previous years.
Response 8:
We adhered to your recommendations and made appropriate adjustments to these aspects as deemed necessary.
Comments 9:
What are visual guidelines?
Response 9:
We have corrected this sentence.
Reviewer 2 Report
Comments and Suggestions for Authors

Author Response
Thank you very much for taking the time to review this manuscript. There were no questions to answer.
Reviewer 3 Report
Comments and Suggestions for Authors
I have read with attention the manuscript titled „ Assessment of Standard Surgical Excision Efficacy and Analysis of Recurrence-Associated Factors in 343 Cases of Nasal Basal Cell Carcinoma: a single-center retrospective study.
The structure of the article is in my opinion very well designed, I appreciate the authors' contribution to the preparation of the article including interesting data on the incidence of BCC.
However, I would suggest some changes.
Shorten the introduction - unnecessary, for example, such detailed information on treatment methods.
In the line 294, a reference needed.
I don't quite understand what the captions under Fig 2 " 2019m, 2020m „ etc. mean?
If there is an analysis of recurrence-associated factors and it must be mentioned that dermoscopy can be useful for the preoperative evaluation of lesions with a higher risk of recurrence.The location of BCC in the H-zone may predispose to its more aggressive course complicated by ulceration and consequently to deeper tissue destruction and the worse prognosis including recurrence.
Therefore,please, consider evaluating findings from the study by Pogorzelska-Dyrbuś J, Salwowska N, Bergler-Czop B. Dermoscopic Pattern of Basal Cell Carcinoma in H- and Non-H-zones. Dermatol Pract Concept. 2023;13(3):e2023125. Published 2023 Jul 1. doi:10.5826/
Author Response
Thank you very much for taking the time to review this manuscript. Please find the detailed responses below and the corresponding corrections highlighted in the re-submitted files.
Comments 1:
Shorten the introduction - unnecessary, for example, such detailed information on treatment methods.
Response 1:
We have corrected the introduction according to your recommendation.
Comments 2:
In the line 294, a reference needed.
Response 2:
We have corrected this line and inserted related references.
Comments 3:
I don't quite understand what the captions under Fig 2 " 2019m, 2020m „ etc. mean?
Response 3:
We have corrected this chart with 2019 year, 2020 year etc.
Comments 4:
If there is an analysis of recurrence-associated factors and it must be mentioned that dermoscopy can be useful for the preoperative evaluation of lesions with a higher risk of recurrence.The location of BCC in the H-zone may predispose to its more aggressive course complicated by ulceration and consequently to deeper tissue destruction and the worse prognosis including recurrence.
Therefore,please, consider evaluating findings from the study by Pogorzelska-Dyrbuś J, Salwowska N, Bergler-Czop B. Dermoscopic Pattern of Basal Cell Carcinoma in H- and Non-H-zones. Dermatol Pract Concept. 2023;13(3):e2023125. Published 2023 Jul 1. doi:10.5826/
Response 4:
Thank you for your valuable insights regarding the advantages of using a dermatoscope. However, it is essential to note that our study is conducted from the standpoint of plastic and reconstructive surgery, where the utilization of a dermatoscope is not part of our standard practice. Patients referred to us from the dermatology department undergo thorough examinations, including dermatoscopy, by specialists in that department.